



# Estimation of Coarse Dead Wood Stocks in Intact and Degraded Forests in the Brazilian Amazon Using Airborne Lidar

Marcos A. S. Scaranello[1], Michael Keller[1,2,3], Marcos Longo[1,3], Maiza N. dos-Santos[1], Veronika Leitold[4], Douglas C. Morton[4], Ekena R. Pinagé[5,6], Fernando Del Bon Espírito-Santo[7,8]

5  [1]Embrapa Informática Agropecuária, Embrapa, Campinas, 13083-886, Brazil
[2]International Institute of Tropical Forestry, USDA Forest Service, Río Piedras, 00926-1119, Puerto Rico
[3]Jet Propulsion Laboratory, California Institute of Technology, Pasadena, 91109, USA
[4]NASA Goddard Space Flight Center, NASA, Maryland, 20771, USA
[5]School of Life Sciences, University of Technology Sydney, Sydney, 2007, Australia
10  [6]College of Forestry, Oregon State University, Corvallis, 97331, USA
[7]Centre for Landscape and Climate Research (CLCR) and Leicester Institute of Space and Earth Observation (LISEO), School of Geography, Geology and Environment, University of Leicester, Leicester, UK
[8]Faculdade de Filosofia, Ciências e Letras de Ribeirão Preto, Universidade de São Paulo, Ribeirão Preto, 14040-900, Brazil

*Correspondence to*: Marcos A. S. Scaranello (masscaranello@gmail.com)



**Abstract.** Coarse dead wood is an important component of forest carbon stocks, but it is rarely measured in Amazon forests and is typically excluded from regional forest carbon budgets. Our study is based on line intercept sampling for fallen coarse dead wood conducted along 103 transects with a total length of 48 km matched with forest inventory plots where standing coarse dead wood was measured in the footprints of larger areas of airborne lidar acquisitions. We developed models to relate lidar metrics and Landsat time series variables to coarse dead wood stocks for intact, logged, and burned or logged and burned forests. Canopy characteristics such as gap area produced significant individual relations for logged forests. For total fallen plus standing coarse dead wood (hereafter defined as total coarse dead wood), the relative root mean square error for models with only lidar metrics ranged from 33% in logged forest to up to 36% in burned forests. The addition of historical information improved model performance slightly for intact forests (31% against 35% relative root mean square error), not justifying the use of number of disturbances events from historical satellite images (Landsat) with airborne lidar data. Lidar-derived estimates of total coarse dead wood compared favorably to independent ground-based sampling for areas up to several hundred hectares. The relations found between total coarse dead wood and structural variables derived from airborne lidar highlight the opportunity to quantify this important, but rarely measured component of forest carbon over large areas in tropical forests.



## 1 Introduction

Intact and disturbed tropical forests play a critical role in the global carbon cycle (Pan et al., 2011). From 1990 through 2007, tropical forests contributed about 46% of the global carbon sink (Schimel et al., 2015). The largest remaining area of tropical forest in the Amazon region contains about 50% of the carbon stored in all tropical forests or 60 Pg-C in the living aboveground

biomass pool (Saatchi et al., 2011; Baccini et al., 2012). The Brazilian Amazon retains about 80% of its original forest cover (PRODES-INPE, 2016) and while deforestation rates in Brazil have decreased by about 70% since 2004 (PRODES-INPE, 2016), forest degradation processes including logging, fire, and fragmentation continue to deplete carbon stocks.

Forest degradation is accelerating the rate of tree mortality across the tropics (McDowell et al., 2018), leading to severe losses

of live aboveground biomass (AGB) (Berenguer et al., 2014; Cochrane, 2003; Longo et al., 2016; Rappaport et al., 2018) and gain of coarse dead wood (CDW) in the forest floor. Aboveground live biomass decreased 35% after logging and 55% after burning + logging in Paragominas Municipality, in the eastern Brazilian Amazon, whereas in Santarem municipality (central Brazilian Amazon) the aboveground live biomass decreased 18%, 17%, 24% after logging, burning, burning+logging, respectively (Berenguer et al., 2014).

In the short term, the stocks of CDW increase substantially after forest disturbance by logging and fire. For example, fallen CDW stocks increased from 55 Mg ha$^{-1}$ in intact forest to 75 Mg ha$^{-1}$ with reduced impact logging, and to almost 110 Mg ha$^{-1}$ in a conventionally logged forest in Paragominas Municipality (Keller et al., 2004). The importance of CDW is magnified in degraded tropical forests (Alamgir et al., 2016). In degraded forests, CDW stocks can exceed the live aboveground biomass

pool (Gerwing, 2002; Palace et al., 2012). Quantifying the spatial and temporal variability of CDW production and decay is therefore critical to constrain the magnitude and timing of carbon emissions from forest degradation or climate anomalies such as droughts (Leitold et al., 2018).

CDW stocks and the rates of decay of CDW constitute large uncertainties in the carbon cycle budget of the Amazon (Aguiar

et al., 2012). We have a limited understanding of how CDW of intact and degraded tropical forests varies across space and time. Traditional forest inventories provide important sources of information for understanding of carbon cycling, but measurements of CDW in tropical forests are rare, labor intensive, and cost prohibitive for large areas (Chao et al., 2009). As an alternative, lidar (light detection and ranging) remote sensing offers the possibility to quantify above-ground biomass (AGB) and CDW over large areas. In contrast to AGB where large number of studies have been developed (e.g. Nelson et al., 1988;

Næsset et al., 2006; Nelson, 2010; Asner et al., 2012; Longo et al. 2016) few studies have focused on lidar remote sensing of CDW and, based on a recent comprehensive review, (Marchi et al., 2018) none has been conducted in intact or degraded tropical forest.



Here, we combine a large dataset of airborne lidar (14,870 ha), Landsat images, and forest inventories of CDW at 14 sites spread across the Brazilian Amazon. Using airborne remote sensing data, we developed the first lidar-derived estimates of CDW for intact and degraded tropical forests including areas that have been logged, burned, and fragmented by deforestation for agricultural expansion.

## 2 Material and methods

### 2.1 Study sites

As part of the Sustainable Landscapes Brazil project, we collected airborne lidar, forest inventories and measures of CDW across five states of the Brazilian Legal Amazon (Para, Amazonas, Mato Grosso, Rondonia, and Acre) (Figure 1). The airborne

lidar data used in this study were collected between 2012 and 2015, covered a total area of 14,870 ha and overlapped with 103 CDW transects (48 km of total length sampled within 6 months of the lidar airborne campaigns). Our sites included two forest types (dense and open evergreen forests) with a moderate climatic variation (precipitation between 1750¬ and 2450 mm yr$^{-1}$), and a large number of disturbance events and processes (Table 1). The dry season length (defined as months with precipitation ≤ 100 mm per month) varies from five months in Tanguro (TAN), Feliz Natal (FNA) and Tapajós (TAP) regions to three

months in Reserva Ducke (DUC). We sampled intact forests as well as forest disturbed by reduced impact logging, conventional logging, understory fire, and combinations of logging and fire. We quantified the number of disturbance events and land-use types using historical Landsat images from between 1984 and 2013 (Longo et al., 2016). We inspected all images using the Normalized Difference Vegetation Index (NDVI) and the Normalized Burn Ratio (NBR). We classified the sites into 5 categories with increasing levels of disturbance: Intact, reduced impact logging, conventional logging, burned, and logged

and burned. We summarized disturbance history by counting the number of degradation events and the time (years) since the last degradation event.

[Figure 1]

[Table 1]

### 2.2 Line intercept sampling of fallen CDW

For this study, we define CDW as material greater than 10 cm in diameter as opposed to fine woody debris (≤ 10 cm) (Harmon et al., 1995). We used the line intercept method for estimating fallen CDW volume (Brown, 1974; Keller et al., 2004; Palace et al., 2007). The line intercept method is a strip sample of infinitesimal width and the data collected in the field are the diameters of wood pieces at their points of intersection with the plane perpendicular to the ground above the line (Brown, 1974). CDW volume was calculated as:

$$V = \left(\frac{\pi^2}{8\,L}\right) \sum D^2$$



Where V is the volume of CDW on an area basis (m3 ha$^{-1}$), D (cm) is the diameter of the wood piece at the line intercept and L (m) is the length of the transect used in sampling (Brown, 1974; Keller et al., 2004). Transect lengths varied from 250 up to 1200 m (16 were 250 m, 86 were 500 m and 1 was 1,200 m). Transects were matched with the inventory plots of living and dead trees and within the coverage area of lidar flights (Figure 1). We used both square plots and belt transects for forest

inventory. When the inventory plot shape was square, four (4) inventory plots were established along the CDW line intercept transect (Figure 1). When the inventory plot was a 20 m wide belt transect, the line intercept transects for fallen CDW sampling bisected the inventory transect. The distance between the transects was at least 50 m in order to maintain independence of the samples based on an estimate of maximum tree height (Keller et al., 2004; Palace et al., 2007). A total of 5 to 22 CDW transects were measured at each site.

We classified the wood pieces in five decomposition classes in the field, following published literature (Harmon et al., 1986; Keller et al., 2004), and converted the volume of CDW into mass by multiplying it by the estimated density of the woody debris. At all sites, the wood density values used were: 0.60, 0.70, 0.58,0.45 and 0.28 Mg m$^{-3}$ for decomposition classes 1, 2, 3, 4 and 5 respectively (1 = intact; 5 = fragmented woody debris) (Keller et al., 2004).

### 2.3 Forest inventory of standing CDW

Several Sustainable Landscapes partners participated in forest inventory so we had three sampling designs. The standing CDW was assessed by using square inventory plots of 40 x 40 m (Sao Felix do Xingu site only) and 50 x 50 m, and also long, narrow belt transects of 20 x 500 m (Longo et al., 2016). All trees above either 5 cm or 10 cm diameter at 1.30 m (DBH) were tagged,

mapped to the nearest 1 m, and diameters were measured using a metric tape with 1 mm resolution (Longo et al., 2016). We used a handheld clinometer and metric tape for field measurements of tree height (Hunter et al., 2013). Snag volume was estimated as a truncated cone using a taper function (Chambers et al., 2000; Palace et al., 2007) for estimating diameter. Volume was converted to mass using the same classes and densities used for fallen CDW.

### 2.4 Lidar data acquisition and processing

Geoid Laser Mapping Ltda. (Belo Horizonte, Brazil) acquired small footprint discrete return lidar (maximum of 4 returns per pulse) during flights in 2012-2014 (Table 1). In 2012 Geoid used an ALTM 3100 (Optech Inc.) while for data acquired in 2013 and 2014 they used a similar ALTM Orion M-200 (Optech Inc.). The height of flights averaged 850-900 m above ground. The field of view was approximately 11º and the line spacing allowed 65% overlap between adjacent swaths. Coverage area per

site varied from 500 to 1996 ha, with mean return density of at least 13 returns m$^{-2}$ (Longo et al., 2016) (Table 1). All transects of fallen CDW and inventory plots were included under the coverage area of lidar flights.



In order to compare lidar metrics to ground based CDW estimates, we established reference polygons using a buffer of 25 m on both sides of the fallen CDW transects. The 50-m total width for our polygons corresponds roughly to the maximum height of a single large tree and was a suitable size to capture canopy gaps. Experiments with narrower transects introduced considerable noise into gap statistics. Wider transects would introduce spatial overlap among samples thereby compromising

the spatial independence of the sample units. Lidar-CDW models were generated and applied at the same resolution (160 x 160 m, or ~25,000 m$^2$).

The lidar point cloud data was processed to produce lidar metrics using the FUSION software (McGaughey, 2014) for all returns (*all-return metrics*) and R environment (R Core Team, 2017) for calculating the metrics when considering only last

laser returns of the forest canopy (*last-return metrics*). The last-return metrics maximize the penetration through the canopy profile and better reflect understory structure (Réjou-Méchain et al., 2015). A digital terrain model (DTM) for each site was supplied by our lidar vendor based on Terrascan software. We previously compared the vendor-provided DTMs with the NASA G-LiHT algorithms (Cook et al., 2013) and field geodesic GNSS measurements and found that they generally agreed to within less than 1-m vertical height (RMSE) at a 1-m horizontal resolution (Leitold et al., 2015). We normalized all

vegetation returns to height above ground by subtracting the height of the DTM at 1-m resolution. We sub-sampled lidar point cloud data by clipping the field plot polygons with the DTM-normalized vegetation returns.

Along with traditional lidar metrics, we also mapped canopy gaps and derived four gap metrics. Forest canopies less than or equal to 10 m in height with a minimum area of 10 m² in the 1-m resolution canopy height model were considered gaps (Hunter

et al., 2015). Gap areas in each plot were summarized based on gap area (m² ha$^{-1}$); mean gap size (m²); standard deviation of gap size assuming a lognormal distribution (m²); and gap count (gaps ha$^{-1}$).

### 2.5 Forest Disturbance History

Based on visual interpretation of Landsat images, we found 30 transects in intact forests, 30 in logged forests, 17 in burned

forests, 14 in logged and burned forests, and 4 in secondary forests (regeneration following complete clearing for agriculture or pasture). For modeling, transects classified as logged and burned were merged into the burned class. Eight transects were not classified because we lacked cloud free images (Figure 1 and Table 1). Where degradation was identified, the number of events ranged from 1 (accounting for 70% of the transects), up to a maximum of 5 in a case where a logging event was followed by 4 events of burning. The median age since the last disturbance was 4 years, ranging from 0.5 (recently logged) up to 23

years following burning.




### 2.6 Statistical models

We developed multivariate linear and nonlinear models relating lidar metrics from a single date acquisition period to CDW. For all models, we summed the fallen CDW from each transect and the mean value of standing CDW from the associated forest inventory belt transect or four square plots (Figure 1), normalized for the area sampled. Through our exploration of the

data, we found no significant general model that applied across all forests and disturbance types. Therefore, we stratified the sites into three classes: intact, logged, and burned. Logged sites included both conventional and reduced impact logging, and burned sites included forests that had been logged and burned. We designated models that used only lidar point cloud metrics as independent variables for a given forest class as lidar-only models. We also developed historical models that included site identity or additional land use history information beyond forest class. The land use history information derived from Landsat

time series included the number of disturbance events and the years since the last disturbance. Detailed information about all Landsat and lidar derived metrics are found in the Table 2. The approaches for model selection for lidar only and historical models are described separately below.

[Table 2]

For lidar-only models, we used the subset selection approach to identify the simplest and most informative combination of

variables (Andersen et al., 2014; Miller, 1984). We excluded highly correlated variables ($r \geq 0.80$) and calculated the variation inflation factor (VIF) in the final models to test for multicollinearity.

For the historical models, we used the framework proposed by (Bolker et al., 2009) for input variable selection. We first selected potential covariates (both Landsat and lidar derived) with expected theoretical relations with CDW. For example, the

age since the last disturbance should be negatively correlated with CDW stocks because of decay (Chambers et al., 2000). After choosing the covariates for logged and burned forests, we fit a full model using ordinary least squares and then performed a backward selection of the best predictors and their combinations using the Bayesian Information Criterion (BIC). For intact forests, we used a mixed effect model including site identity as both a fixed and random variable (Pinheiro and Bates, 2000).

We log transformed (natural log) the response variables when necessary for improved model prediction and error distribution assumptions. We then back-transformed using the Baskerville bias corrector ($\exp(\sigma_\varepsilon^2/2)$) for model assessment (Baskerville, 1972). We used adjusted $R^2$, relative bias (bias, in %; mean error divided by observed mean) and relative root mean square error (RMSE, in %; square root of the mean squared error divided by the observed mean) as goodness-of-fit measures for comparison with other studies on lidar-CDW models (Pesonen et al., 2008). We did not calculate adjusted $R^2$ for the linear

mixed effect model because of the difference in accounting for the number of parameters in both fixed and random terms, compared to the ordinary least squares method (Bolker et al., 2009).





## 3 Results

### 3.1 Field sample CDW variability

The overall mean (±standard deviation) total CDW (including fallen and standing dead wood) stock grouped by site was 50.6 (±17.7) Mg ha$^{-1}$. Individual site averages ranged from 21.8 Mg ha$^{-1}$ to 93.0 Mg ha$^{-1}$ (Table 3). When grouped by degradation

level and site, average total CDW was lower for burned forests (40.4 ±29.7 Mg ha$^{-1}$) than logged forests (70.9 ± 19.9 Mg ha$^{-1}$). In comparison, intact forests grouped by site had average total CDW of 42.4 (±19.7) Mg ha$^{-1}$. Logged forests had the largest CDW stocks with average of 70.9 ±19.9 Mg ha$^{-1}$, and recorded the largest CDW stock (150 Mg ha$^{-1}$) in a single transect in FST, where logging had occurred less than 6 months prior to the data collection. The mean total CDW stock was 21.0 (± 2.0) Mg ha$^{-1}$ in TAN intact transects, less than DUC and TAP intact transects. The mean total CDW stock was 57.6 (± 15.0) Mg

ha$^{-1}$ and 66.2 (± 20.0) Mg ha$^{-1}$ in DUC and TAP, respectively.

[Table 3]

### 3.2 Modeling scenarios

The best lidar-based predictor of total coarse wood debris for transects classified as intact was the 75$^{th}$ percentile of last returns

(m) (Figure 2a). The gap area (m$^2$ ha$^{-1}$) was the best predictor of total coarse wood debris for transects classified as logged (Figure 2b). For burned forests, total CDW was inversely related to the return fraction above 30 m (Figure 2c).

[Figure 2]

Models for total CDW in the *lidar only* scenarios generally performed well (Table 4; Figure 3). Relative RMSE ranged from 33% for total CDW in logged forest to up to 36% in burned forest (Table 4). The predictions depended, in part, on last return

metrics for intact forest classes and notably, gap area for the logged class. The 1$^{st}$ and 10$^{th}$ percentile of all returns, as well as mode of all return heights, were also important for predictions in the logged and burned classes. Models that separately considered fallen and standing CDW components produced poorer fits than models of total CDW (Table C2).

[Table 4]

[Figure 3]

The inclusion of disturbance history and site identity in the *historical models* led to modest improvements on the quality of prediction for total CDW in intact, a very small gain (1% decrease in RMSE) for logged forests and a poorer fit (9% increase in RMSE) for burned forest (Table 5). The *historical model* for intact forest included two site related variables in the mixed model, a site factor and a random slope for canopy relief ratio in each site. Historical models that separately fit fallen and standing CDW components produced poorer results than for total CDW (Table C3).

[Table 5]

Although the *lidar only* models had a relatively good performance measured by adjusted R² and RMSE, we also examined whether the models were biased. In general, we found no evidence of biases (Figures 4 and 5) or heteroskedasticity in the




model residuals (Figure 5). Measured by mean relative bias, the model for burned forests had the poorest performance among the lidar-only models with a value of -3%. The mean relative bias for the *historical* scenario were 0.0%, 0.0% and -3.9% for the intact, logged and burned forests respectively (Table 5).

[Figure 4]

### 3.3 Landscape level prediction of CDW

For comparison to published field surveys, we applied the *lidar-only* models over the entire lidar scenes (~1000 ha each) for three intact sites, one logged site and one burned site at a 166 m resolution (Figure 5). For the Tapajós National Forest intact site (TAP), the landscape level predicted mean was $51.3 \pm 18.8$ (standard deviation) Mg ha$^{-1}$ and the range was 15-91 Mg ha$^{-1}$

after excluding one outlier pixel located on the edge of the lidar scene with 146.0 Mg ha$^{-1}$ of CDW (Figure 5a). For the Reserva Adolpho Ducke intact site (DUC), the landscape mean CDW was $41.6 \pm 5.0$ Mg ha$^{-1}$ and the range was 22-61 Mg ha$^{-1}$ (Figure 5b). For the Fazenda Tanguro intact site (TAN), the landscape mean CDW was $21.0 \pm 2.0$ Mg ha$^{-1}$ (Figure 5c).

For the Fazenda Cauaxi logged site (CAU), the landscape mean CDW was $84.6 \pm 27.5$ Mg ha$^{-1}$ and the landscape mean for intact forests at the same site was $54.2 \pm 8.8$ (Figure 5d) Mg ha$^{-1}$. At this site, in the logged forests there were extremely high

predicted values ranging from 161.0 Mg ha$^{-1}$ to up to 200.0 Mg ha$^{-1}$ (Figure 5d). The occurrence of gap areas out of the range used for calibration contributed to the prediction of those outliers. Finally, for the burned site in Fazenda Tanguro the predicted landscape level mean of 46.5 Mg ha$^{-1}$ was about twice the mean for undisturbed forest at this site (Figure 5c).

[Figure 5]

## 4 Discussion

### 4.1 Lidar models and controls on CDW

Necromass stocks in intact forests are controlled by the balance between inputs from tree and branch fall and loss from CDW decay (Chao et al., 2009; Palace et al., 2008). The slight increase in the performance of the model for intact forests in the *historical* scenario (Table 4 and 5), compared to the *lidar only* model, highlights that differences in site-specific characteristics

controlling the input and decay of CDW might be important for predicting CDW in Amazonian forests. For the *lidar-only* scenario we found an increase in total CDW in the intact forests with increasing values of the 75$^{th}$ percentile of last returns, a metric related to the overall increase in both canopy and understory height and a correlate of total biomass. Our results are consistent with Chao et al. (2009), who also found a weak correlation between total CDW and live biomass whereas Martins et al. (2015) related CDW stocks with mean biomass per tree.





Logging and fire differentially affected CDW in Amazonian degraded forests. Fire events tended to produce more standing CDW than fallen, whereas most of the total CDW in logged forest was fallen (data not shown). The ratio between standing and fallen CDW is suggestive of the predominant mode of tree death. The most pronounced difference between logging and fire was the effect of gap area on the amount of CDW (Figure 2). A significant amount of CDW is associated with gap creation

in intact Amazonian forests (Espírito-Santo et al., 2014a, 2014b) and our models for logging confirm a strong relation between gap area and CDW. Considering both age since the last disturbance and number of degradation events in the *historical* models, gap area was still positively related to CDW stocks in logged forests whereas the opposite trend was found in burned forests. For example, for a single event of one-year age in the *historical* models, the increase of gap area from 300 m² ha$^{-1}$ to up to 1000 m² ha$^{-1}$ led to an increase of 0.06 Mg of CDW per m² of gap in logged forests and a decrease of 0.012 Mg of CDW per

m² of gap in burned forests. For additional fire events, there are compensatory effects controlling CDW stocks. Fires lead to mortality, thereby increasing stocks, but also consume existing CDW at the time of the fire. The opposite signs of the parameters for gap area and number of events reflect these opposing controls.

### 4.2 Comparisons to other lidar-related models

Two classes of models have been used for CDW estimation using lidar: (1) Area-based models estimate CDW indirectly based on lidar metrics calibrated with data from forest inventory plots (Martinuzzi et al., 2009; Pesonen et al., 2008);  and (2) individual-based models that identify standing dead trees (Casas et al., 2016) and downed trees on the ground (Blanchard et al., 2011; Polewski et al., 2015). The individual-based approach is generally more appropriate for identifying and estimating volume or basal area of standing and fallen dead trees in more open canopies, and lack of dense vegetation, compared to the

dense tropical forests that we studied (Blanchard et al., 2011). We employed only area-based models and so we will compare our results only to other results of this category.

In the area-based approach, CDW metrics may reflect underlying mechanisms generating CDW. For example, lidar metrics related to gaps such as intensity of returns accumulated closer to the ground and standard deviation of returns were both

included as predictors of fallen CDW volume in a boreal forest from Eastern Finland (Pesonen et al., 2008). The model for fallen coarse woody volume had a relative RMSE of 51.6%, is similar to the performance of the model for burned sites in our historical scenario. As we found, in the area-based approach, models for predicting standing necromass are poorer than the models for fallen woody debris similar to findings in boreal forest (Pesonen et al., 2008). The boreal forest model for standing dead tree volume had a relative RMSE of 78.8% (Pesonen et al., 2008).

Our landscape level means and ranges at the four intact sites, as well as at the logged and burned site, were similar to published field surveys. Hayek et al. (2018) found 61.0 ± 14.8 Mg ha$^{-1}$of CDW stock in the Tapajós National Forest (51.3 ± 18.8 Mg ha$^{-1}$ from this study). Martins et al. (2015) reported a range of 6.7-72.9 Mg ha$^{-1}$ of CDW stock in the Reserva Adolpho Ducke



(mean of 41.6 and range of 22-60 Mg ha$^{-1}$ from this study). Keller et al. (2004) found an average of 55.2 ± 4.7 Mg ha$^{-1}$ and 74.7 ± 0.6 Mg ha$^{-1}$ of fallen CDW in an intact and logged site respectively at Fazenda Cauaxi (54.2 ± 8.8 Mg ha$^{-1}$ and 84.6 ± 27.5 Mg ha$^{-1}$ from this study).

Preliminary analysis of wall-to-wall maps created with our lidar-only models alongside the histograms (Figure 5) revealed a unique potential for explaining spatial patterns of CDW in intact forests and assessing the effect of degradation on CDW stocks. The total CDW stock was higher in eastern and central Amazonian intact forests than in the southern intact forests (Chao et al., 2009). In addition, the spatial pattern and the dispersion of CDW distribution in TAP and DUC sites illuminates the mechanisms controlling CDW at landscape level. First, the CDW stocks at DUC are strongly related to topography (Figure

B1). At DUC site, there is a pattern of higher stocks of CDW in the plateau, where the soil is more structured, deeper and less physically restricted and where AGB stocks are greater (Martins et al 2014). On the other hand, at TAP site, the peaks of CDW stocks appears to be more spatially disaggregated which might indicate that CDW is associated with natural, small-scale natural disturbance (Rice et al., 2004). For degraded forest, in the Fazenda Tanguro site we found no published data on CDW but the increase in CDW after the repeated fire events agree with previous studies (Gerwing, 2002). Finally, the effect of logging (1.5-

fold increase) on CDW stocks at the landscape level at CAU site similar in magnitude to our earlier field studies (Keller et al. 2004).

### 4.3 Implications for studies of the Amazon carbon budget

Our results demonstrate that small footprint airborne lidar remote sensing can be used to reduce uncertainty of the spatial

distribution of CDW stocks across intact and degraded Amazonian forests. Our approach required systematic classification of the Amazonian forests into intact, logged, and burned conditions. More complex models using regression trees may eventually combine classification and CDW estimation using lidar data. We avoided more complex models in this study because our simple regression models are more transparent and less likely to suffer from overfitting because they rely on few predictors. Models for estimation of CDW using lidar data only are likely to be less accurate than models for total above-ground biomass

(AGB) when relative uncertainty is compared (e.g. Longo et al. 2016). However, because the absolute values of CDW are usually in the range of 10 to 20% of AGB except at heavily degraded sites, the absolute uncertainties for CDW are still likely to be smaller than the absolute uncertainties for AGB.

For any extrapolation approach, it is critical to avoid bias. Overall, we found little bias in our models for estimation of CDW

across forest sites and disturbances types. Nonetheless, we raise two potential concerns. First, in intact forests of the southern Amazon the stocks of CDW are considerably lower than central and eastern Amazon. This reflects the smaller biomass stocks and lower wood densities found in that region (Nogueira et al., 2007).  Second, in heavily burned forests (more than 3 events of fire) the in-situ estimates of CDW stocks were well below the airborne-lidar predicted values, probably because CDW was

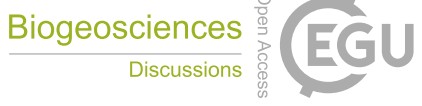

consumed in the fires. We note that forest degradation from repeated fires is concentrated along the eastern edge of the Brazilian arc of deforestation (Morton et al., 2013).

Improved knowledge of the spatial distribution of CDW stocks complementing our growing knowledge of aboveground live

biomass distributions will reduce the uncertainties of emissions from deforestation and forest degradation (Aguiar et al. 2012). We highlight that CDW is relatively more abundant in degraded than in intact forests. Airborne lidar is a valuable tool for estimates of the impact of forest degradation on the carbon cycle, and our work has the potential to expand understanding beyond the current lidar approaches that focus exclusively on aboveground biomass. Further development of the approach presented here may be applied to more extensive and systematic airborne lidar acquisitions or perhaps even spaceborne lidar

from GEDI and/or ICESat-2 missions to estimate CDW across wide areas of tropical forests (Stavros et al., 2017).

**Author contribution**

M. A. S. Scaranello and Michael Keller designed the study. Michael Keller and Maiza N. Dos-Santos coordinated the field and lidar campaigns and Maiza N. Dos-Santos analyzed the field data. M. A. S. Scaranello analyzed the field and lidar data

and conducted the modeling analysis. Marcos Longo provided the data on visual interpretation of Landsat imagery and assisted with modeling analysis. Douglas C. Morton participated in the modeling analysis. Veronika Leitold and Ekena R. Pinagé provided data on visual interpretation of Landsat imagery. Fernando D. B. Espírito-Santo provided the Tapajós and Reserva Ducke field data. M. A. S. Scaranello and Michael Keller wrote the paper. All co-authors revised and approved the paper.

**Competing interests**

We declare that we have no conflict of interests.

**Data availability**

Field coarse woody debris and forest inventory data as well as lidar data are available from EMBRAPA at the following URL:

https://www.paisagenslidar.cnptia.embrapa.br/webgis/

**Acknowledgements**

Data were acquired by the Sustainable Landscapes Brazil project implemented by the Brazilian Agricultural Research Corporation (EMBRAPA) and the US Forest Service with financial support from USAID and the US Department of State;

EMBRAPA; CNPq LBA project (457927-2013-5); CNPq CSF project (457927/2013-5); NASA's Carbon Monitoring System. M. Keller was supported in part by the Next Generation Ecosystem Experiments-Tropics, funded by the US Department of Energy, Office of Science, Office of Biological and Environmental Research.; M. Scaranello was supported by CNPq (fellowship grant 384475/2015-9). M. Longo was supported by FAPESP (grant 2015/07227-6). F. Espírito-Santo was supported by Natural Environment Research Council (NERC) grants ('BIO-RED' NE/N012542/1 and 'AFIRE'



NE/P004512/1) and Newton Fund ('The UK Academies/FAPESP Proc. N°: 2015/50392-8 – Fellowship and Research Mobility'). D. Morton was supported by CNPq (grant 457927/2013-5 and the Ciência sem Fronteiras Program) and NASA's Carbon Monitoring System.

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



**Figure captions**

**Fig. 1.** Location of the study sites in the States of Brazilian Legal Amazon. Site codes are shown at approximate locations (see Table 1). The lower inset shows the canopy height model (m) scene from AND site as an example of the lidar data covering the field transects for sampling standing and fallen dead wood. The upper inset shows the sample design used with line intercept

samples to quantify fallen CDW and associated square forest inventory plots for aboveground live and standing CDW. . CAU, FST, JAM, TAL, TAN, DUC and TAP sites were classified as intact; CAU, FST, JAM sites were classified as reduced impact logging; BON, PAR, BET sites were classified as conventional logging; AND, HUM, BET, SFX, TAL and TAN sites were classified as burned; AND, BON, FNA and PAR sites were classified as logged and burned; PAR, BET, SFX sites were classified as secondary; BON, CAU, HUM, SFX, and TAN sites had transects unclassified.

**Fig. 2.** Relationship between total CDW (TCDW) and the best single lidar-based predictor variable of TCDW: (a) the 75th percentile of last returns (m) (n=30) for transects classified as intact; (b) gap area (m² ha⁻¹) (n=23) for reduced impact logging transects; (c) return fraction above 30 m (last returns) for transects classified as burned (n=30). (a) ln(TCDW) = 1.96 + 0.08·75$^{th}$ percentile of last returns. P<0.01, Adjusted $R^2$: 0.36; (b) TCDW = 1.04·gap area$^{0.66}$. P<0.01, Adjusted $R^2$: 0.57; (c) ln(TCDW) = -3.97-11.95·return fraction above 30 m. P<0.01, Adjusted R2: 0.25.

**Fig. 3.** Measured values of total CDW (TCDW) versus values predicted by the models for *lidar only* scenarios for forests classified as (a) intact (Adjs-$R^2$: 0.44; RMSE (%): 35.1), (b) logged (Adjs-$R^2$: 0.50; RMSE (%): 33.0) and (c) burned (Adjs-$R^2$: 0.51; RMSE (%): 36.0).

**Fig. 4.** Residuals versus predicted values of total CDW (TCDW) by the models for *lidar only* scenario for forests classified as (a) intact (mean bias (%): -0.41), (b) logged (mean bias (%): 0.00) and (c) burned (mean bias (%): -3.00).

**Fig. 5.** Wall-to-wall maps and histograms of total coarse wood debris predicted by lidar only models at landscape level (166-m resolution) for intact forest at the Tapajós National Forest (a) the predicted mean was 51.3 ± 11.8 Mg ha⁻¹ (red dotted line) and the field-based mean from our database was 66.2 ± 20.0 Mg ha⁻¹ (black dotted line). For intact forest at Reserva Adolph Ducke (b) the predicted mean was 41.6 ± 5.0 Mg ha⁻¹ and the field-based mean from our database was 57.6 ± 20.0 Mg ha⁻¹. For intact forests at Fazenda Tanguro (c) the predicted mean was 21.0 ± 2.0 Mg ha⁻¹ and field-based mean was 20.6 ± 1.8 Mg

ha⁻¹. For burned forests (highlighted as a second panel in the CDW map) the predicted mean was 46.5 ± 9.8 Mg ha⁻¹ and the field-based mean was 32.5 ± 6.0 Mg ha⁻¹. For intact forests at Fazenda Cauaxi (d) the predicted mean was 54.2 ± 8.8 Mg ha⁻¹ and field-based mean was 33.2 ± 10.0 Mg ha⁻¹. For logged forests the predicted mean was 84.6 ± 27.5 Mg ha⁻¹ and the field-based mean was 60.3 ± 24.0 Mg ha⁻¹; high CDW areas in the norther portion of the image are associated with the main road through the logging site.

**Figures**

**Fig. 1.**

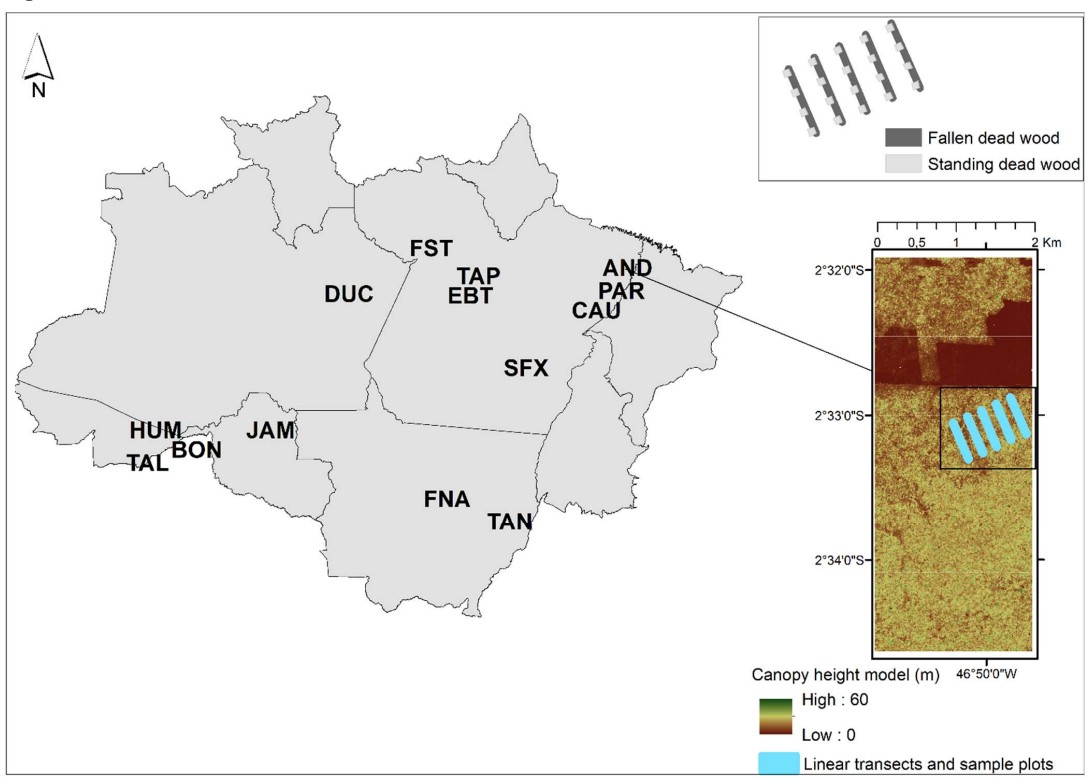



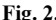

**Fig. 2.**

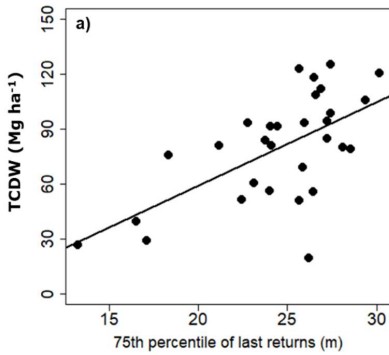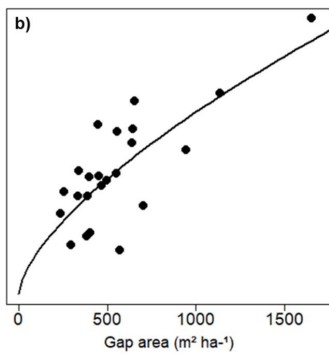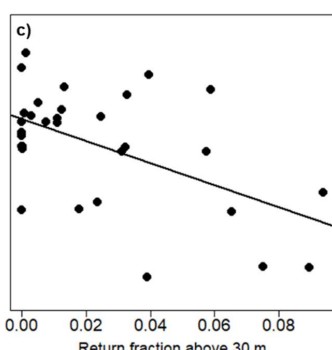



**Fig. 3.**

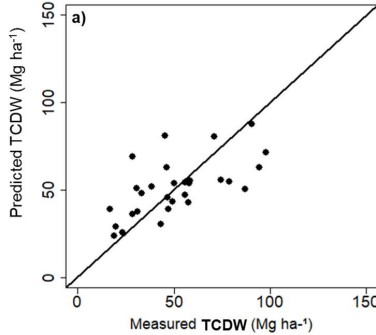 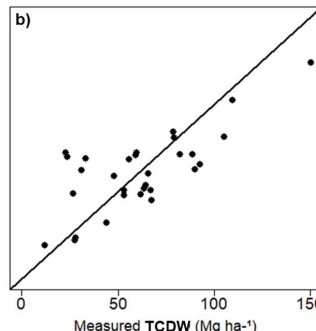 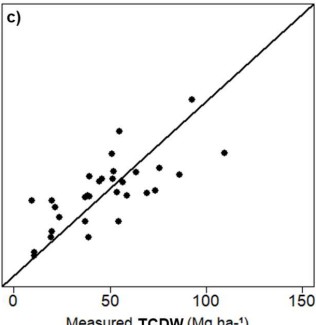



**Fig.4.**

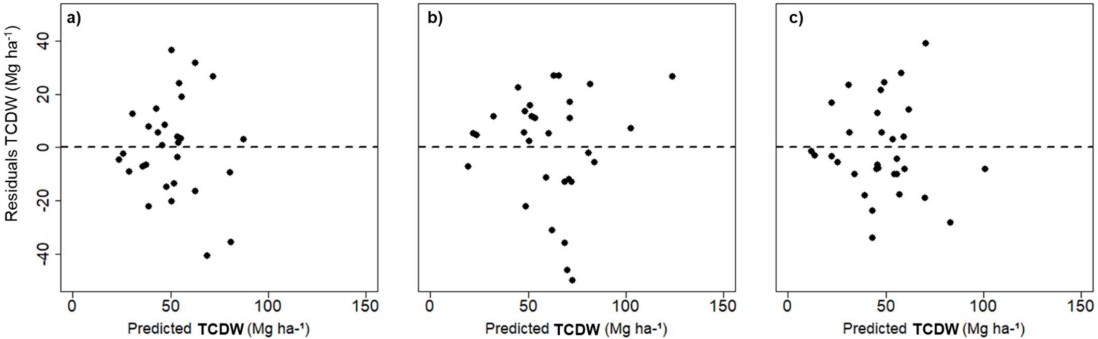



**Fig. 5.**

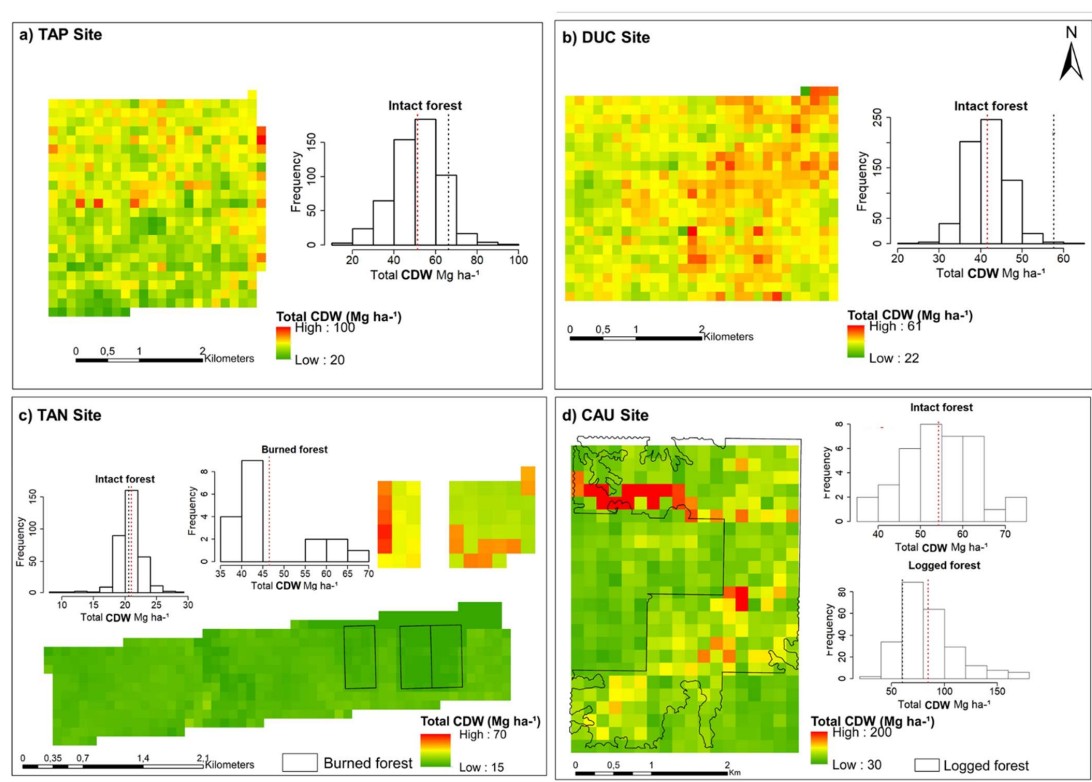



**Table captions**

**Table 1.** Description and location of study sites. Forest status identifies degradation classes, where BRN=burned; CVL= conventional logging; CVL+BRN=logged and burned; INT=intact; RIL=reduced impact logging; SEC=second growth; and UKN=unclassified.

**Table 2.** Landsat-derived variables and lidar metrics used as potential covariates for modeling CDW in intact and degraded Amazonian forests.

**Table 3.** CDW (mean and standard deviation) by degradation level and site. AGB, standing CDW, fallen CDW and total CDW are in Mg ha$^{-1}$.

**Table 4.** Equations, adjusted R², mean relative bias (%) and relative root mean square error (RMSE in %) of the *lidar only* scenario for predicting CDW in intact and degraded forests using lidar variables. $rf_{0\ 1m}$ is return fractions between 0 and 1 m height of the *last returns*; $P75_{last}$ is the 75$^{th}$ percentile of *last returns* in meters; gap area is gap area in m$^2$ ha$^{-1}$; Mode_all is mode of *all returns* in meters. $P05_{all}$ is the 5$^{th}$ percentile of *all returns* in meters; $rf_{above\ 30m}$ is return fraction above 30 meters of *all returns*. EN is residual following a normal distribution with μ and σ.

**Table 5.** Equations, adjusted R², mean relative bias (%) and relative root mean square error (RMSE in %) of the *historical scenario* for predicting CDW in intact and degraded forests using Landsat and lidar variables. The parameters of the mixed-effect model for intact forests are shown in the Table C1 as appendices. Age is the number of years since the last disturbance event. Gap Area is total gap area in m² ha$^{-1}$. $P05_{last}$ is the 5$^{th}$ percentile of the *last returns*. Number Event is the count of degradation events. CRR is canopy relief ratio. EN is residual following a normal distribution with μ and σ. Estimated parameters by each site (fixed effect) are in the Table C1.





**Tables**

**Table 1.**

| Site Identity | Region and state | Long | Lat | Annual rainfall (mm) | Year of field sampling | Transect length (m) | Forest status | Airborne lidar survey | | |
|---|---|---|---|---|---|---|---|---|---|---|
| | | | | | | | | Date | Area (ha) | Return density (m⁻²) |
| AND | Paragominas, Pará | 46.83 W | 2.55 S | 2181 | 2013 | 500 | BRN (4), CVL+BRN (1) | 2014 | 1000 | 38.2 |
| BON | Rio Branco, Acre | 67.29 W | 9.87 S | 2017 | 2013 | 250 | CVL (2), CVL+BRN (3), UKN (1) | 2013 | 600 | 33.4 |
| CAU | Paragominas, Pará | 48.48 W | 3.75 S | 2180 | 2012 | 500 | INT (4), RIL (16), UKN (2) | 2012 | 1214 | 28.3 |
| DUC | Reserva Ducke, Amazonas | 59.94 W | 2.95 S | 2404 | 2012 | 500 | INT (10) | 2012 | 1248 | 22.7 |
| FNA | Feliz Natal, Mato Grosso | 55.01 W | 12.50 S | 1812 | 2013 | 500 | CVL+BRN (5) | 2013 | 1200 | 38.3 |
| FST | Saracá-Taquera, Pará | 56.22 W | 1.62 S | 2429 | 2013 | 500 | INT (1), RIL (4) | 2013 | 1021 | 32.9 |
| HUM | Rio Branco, Acre | 67.65 W | 9.76 S | 2012 | 2013 | 250 | BRN (3), UKN (3) | 2013 | 501 | 66.6 |
| JAM | Jamari N. Forest, Rondônia | 63.01 W | 9.12 S | 2054 | 2011, 2013 | 500 | INT (3), RIL (3) | 2013 | 1673 | 31.0 |
| PAR | Paragominas, Pará | 47.53 W | 3.32 S | 1817 | 2013 | 500 | CVL (4), CVL+BRN (5), SEC (1) | 2014 | 1003 | 40.0 |
| EBT | Belterra, Pará | 54.80 W | 3.21 S | 2098 | 2014 | 500 | CVL (1), BRN (1), SEC (2) | 2014 | 850 | 49.5 |
| SFX | São Félix do Xingu, Pará | 52.00 W | 6.50 S | 2100 | 2012 | 250-1200 | BRN (4), UKN (1), SEC (1) | 2012 | 1996 | 30.1 |
| TAL | Rio Branco, Acre | 67.98 W | 10.26 S | 1980 | 2014 | 250 | INT (1), BRN (2) | 2014 | 500 | 40.7 |
| TAN | Fazenda Tanguro, Mato Grosso | 52.41 W | 13.08 S | 1767 | 2012 | 500 | INT (2), BRN (3), UKN (1) | 2012 | 1006 | 13.1 |
| TAP | Tapajos National Forest, Pará | 54.95 W | 2.86 S | 2030 | 2012 | 500 | INT (9) | 2012 | 1049 | 25.1 |




**Table 2.**

| Landsat | Description |
|---|---|
| Degradation class | Status of degradation such as intact, logged, burned and burned after logging. |
| Age since the last degradation event | The age (years) since the last degradation event. |
| Number of degradation events | Number of events of logging or burning. |
| **Lidar** | |
| Percentiles | Percentiles 01, 05, 10, 20, 30, 40, 50, 60, 70, 80, 90 and 99 of the return distribution. |
| Return fraction among height intervals | Fractions of returns among pre-determined height intervals (e.g. from 0 to 1 m) or above a pre-determined height (e.g. above 20 m). |
| Gap metrics | Mean gap size (m²), standard deviation of gap size, standard deviation of gap size assuming a log-normal distribution, gap area (m² ha$^{-1}$) and gap count (gaps ha$^{-1}$) |
| Canopy Relief Ratio (Parker et al. 2004) | A quantitative descriptor of the relative shape of the canopy defined as: ((Mean height—Min height) / (Max height—Min height)) |
| Moments of return distribution | Mean, median, variance, skewness and kurtosis |
| L-moments of return distribution | L moments (1$^{st}$, 2$^{nd}$, 3$^{rd}$ and 4$^{th}$) are linear combinations of ordered data values (elevation returns) described by (Hosking, 1990), analogous to traditional moments. |





**Table 3.**

| Site | Degradation level | n | AGB mean | sd | Standing CDW mean | sd | Fallen CDW mean | sd | Total CDW | CDW/AGB |
|------|------------------|---|----------|-----|-------------------|-----|-----------------|------|-----------|---------|
| AND | BRN | 4 | 184.0 | 21.9 | 17.7 | 5.0 | 40.6 | 10.4 | 58.2 | 0.3 |
|  | CVL+BRN | 1 | 106.4 | - | 21.7 | - | 54.1 | - | 75.8 | 0.7 |
| BON | CVL | 2 | 211.9 | 49.9 | 5.2 | 6.1 | 14.7 | 4.8 | 19.9 | 0.1 |
|  | CVL+BRN | 3 | 166.2 | 10.6 | 5.7 | 3.8 | 38.8 | 9.3 | 44.5 | 0.3 |
|  | UKN | 1 | 342.7 | - | 0.7 | - | 7.2 | - | 7.9 | 0.0 |
| CAU | INT | 4 | 410.6 | 13.7 | 7.1 | 5.3 | 26.1 | 10.4 | 33.2 | 0.1 |
|  | RIL | 16 | 351.7 | 58.8 | 12.6 | 6.5 | 47.8 | 24.2 | 60.3 | 0.2 |
|  | UKN | 2 | 489.7 | 63.0 | 15.6 | 5.6 | 22.1 | 6.3 | 37.7 | 0.1 |
| DUC | INT | 10 | 325.3 | 129.6 | 9.2 | 6.0 | 48.5 | 15.7 | 57.6 | 0.2 |
| FNA | CVL+BRN | 5 | 3.8 | 4.7 | 18.3 | 4.7 | 29.3 | 6.4 | 47.6 | 12.6 |
| FST | INT | 1 | 416.9 | - | 25.7 | - | 37.1 | - | 62.9 | 0.2 |
|  | RIL | 4 | 345.6 | 66.8 | 16.7 | 8.7 | 83.8 | 41.6 | 100.5 | 0.3 |
| HUM | BRN | 3 | 194.6 | 44.7 | 2.5 | 0.8 | 10.5 | 5.9 | 13.1 | 0.1 |
|  | UKN | 3 | 192.7 | 97.6 | 5.1 | 3.7 | 49.4 | 12.5 | 54.5 | 0.3 |
| JAM | INT | 3 | 264.2 | 45.0 | 16.8 | 1.2 | 19.7 | 10.6 | 36.5 | 0.1 |
|  | RIL | 3 | 228.1 | 78.1 | 20.3 | 4.0 | 40.2 | 8.2 | 60.4 | 0.3 |
| PAR | CVL | 4 | 151.6 | 48.6 | 9.5 | 2.5 | 37.9 | 25.1 | 47.4 | 0.3 |
|  | CVL+BRN | 5 | 81.2 | 22.7 | 17.8 | 8.0 | 55.3 | 18.9 | 73.1 | 0.9 |
|  | SEC | 1 | 109.6 | - | 8.2 | - | 35.7 | - | 43.8 | 0.4 |
| EBT | CVL | 1 | 155.1 | - | 13.5 | - | 53.8 | - | 67.3 | 0.4 |
|  | BRN | 1 | 129.3 | - | 30.2 | - | 62.5 | - | 92.7 | 0.7 |
|  | SEC | 2 | 107.4 | 17.4 | 5.8 | 0.9 | 32.6 | 12.5 | 38.4 | 0.4 |
| SFX | BRN | 4 | 199.4 | 71.6 | 7.6 | 3.2 | 11.4 | 4.1 | 19.0 | 0.1 |
|  | UKN | 1 | 155.5 | - | 30.1 | - | 13.0 | - | 43.1 | 0.3 |
|  | SEC | 1 | 145.0 | - | 34.1 | - | 17.9 | - | 52.0 | 0.4 |
| TAL | INT | 1 | 150.7 | - | 6.2 | - | 12.7 | - | 18.9 | 0.1 |
|  | BRN | 2 | 138.5 | 5.5 | 12.0 | 1.5 | 42.1 | 19.8 | 54.1 | 0.4 |
| TAN | INT | 2 | 167.1 | 38.9 | 10.4 | 1.8 | 11.3 | 0.8 | 21.6 | 0.1 |
|  | BRN | 3 | 160.1 | 26.0 | 19.9 | 9.0 | 12.7 | 6.0 | 32.5 | 0.2 |
|  | UKN | 1 | 114.1 | - | 21.3 | - | 11.3 | - | 32.5 | 0.3 |
| TAP | INT | 9 | 247.7 | 101.4 | 6.6 | 5.2 | 59.6 | 21.9 | 66.2 | 0.3 |

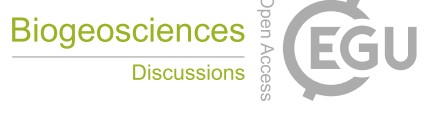

**Table 4.**

| Land use | Equation | Adjs-$R^2$ | Bias (%) | RMSE (%) |
|---|---|---|---|---|
| Intact | $\ln TCWD = 1.00(0.33)\, rf_{01m} - 0.07(0.03)\, P75_{last}^{\,0.34(0.11)} + EN(\mu=0,\ \sigma=0.36)$ | 0.44 | -0.41 | 35.1 |
| Logged | $TCWD = -48.63(22.27) + 0.07(0.01)\, gap\,area + 2.40(0.63)\, Mode\_all + 190.98(83.26)\, P01_{all}$ $+ EN(\mu=0,\ \sigma=22.02)$ | 0.50 | 0.00 | 33.0 |
| Burned | $\ln TCWD = 3.88(0.14) + 1.05(0.27)\, P05_{all} - 0.03(0.01)\, Mode_{all} - 11.91(2.05)\, rf_{above\,30m}$ $+ EN(\mu=0,\ \sigma=0.45)$ | 0.51 | -3.00 | 36.0 |



**Table 5.**

| Land use | Equation | Adjs-R² | Bias (%) | RMSE (%) |
|---|---|---|---|---|
| | Fixed effect variables     Random slope by site | | | |
| Intact | $TCWD = 3.40 + Site\ Factor + 119.72(126.18)CRR + \quad CRR\ by\ site \quad + EN(\mu = 0,\ \sigma = 14.14)$ | - | 0.00 | 31.3 |
| Logged | $TCWD = 26.29(13.71) - 5.45(1.62)\ Age + 0.06(0.01)\ Gap\ Area + 65.50(20.71)\ P05_{last}$ $+EN(\mu = 0,\ \sigma = 20.43)$ | 0.52 | 0.00 | 32.0 |
| Burned | $\ln TCWD = 3.43(0.24) - 0.05(0.01)\ Age - 0.0003(0.00)\ Gap\ Area + 0.53(0.10)\ Number\ Event$ $+EN(\mu = 0,\ \sigma = 0.46)$ | 0.46 | -3.9 | 45.0 |





**Appendices**

**Figure B1.** Wall-to-wall map of total CDW and digital terrain model at DUC site.

### Reserva Ducke

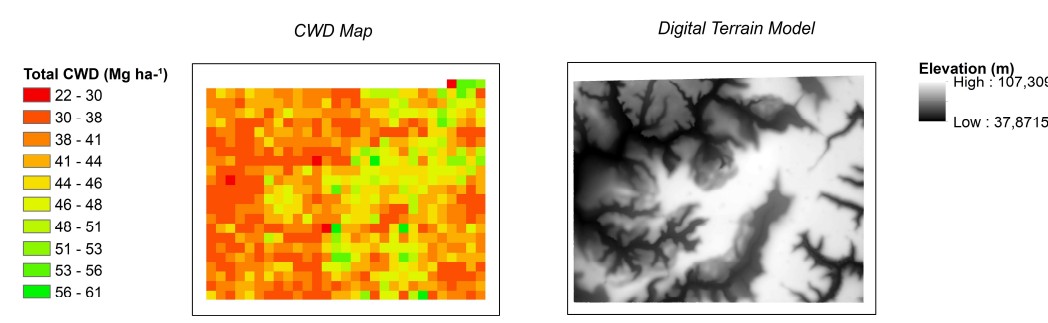





**Table C1.** Fixed and random effects parameters of the *historical scenario* model for intact forests.

| Parameter (fixed effect) | Estimate | Standard deviation |
|---|---|---|
| Intercept | 3.4 | 64.3 |
| Site DUC | 49.7 | 68.4 |
| Site FST | 10.4 | 100.3 |
| Site JAM | 1.3 | 91.4 |
| Site TAL | -30.9 | 96.6 |
| Site TAN | -35.6 | 104.6 |
| Site TAP | -63.5 | 74.9 |
| Canopy relief ratio | 119.7 | 126.2 |
| **Random term (canopy relief ratio by site, see Table 5)** | **Estimate** | |
| Site CAU | 75.2 | |
| Site DUC | 9.9 | |
| Site FST | 119.7 | |
| Site JAM | 79.6 | |
| Site TAL | 119.7 | |
| Site TAN | 109.7 | |
| Site TAP | 324.0 | |



**Table C2**. Selected variables (bold indicates positive signal of parameter), adjusted R², mean relative bias (%) and relative root mean square error (RMSE in %) of the *lidar only* scenario for estimating fallen and standing CDW in intact and degraded forests using lidar variables.

| Land use class | Lidar only scenario | | | | | | | |
|---|---|---|---|---|---|---|---|---|
| | Fallen CDW | | | | Standing CDW | | | |
| | Predictors | Bias (%) | R² | RMSE (%) | Predictors | Bias (%) | R² | RMSE (%) |
| Intact | **P75 last returns,** return fraction 0-1 m | -2.52 | 0.51 | 41.3 | **Gap count,** Gap count² | -8.50 | 0.18 | 56.0 |
| Logged | SD gap size, interquartile range last returns, P10 last returns | 0.00 | 0.66 | 32.4 | - | | - | - |
| Burned | P10 last returns, **mode all returns, return fraction 25-30 m** | 6.00 | 0.39 | 48.0 | P10 last returns, 3th L moment all returns, **mode all returns** | 7.31 | 0.43 | 47.4 |





**Table C3.** Selected variables (bold indicates positive signal of parameter), adjusted R², mean relative bias (%) and relative root mean square error (RMSE in %) of the *historical scenario* for estimating fallen and standing CDW in intact and degraded forests using Landsat and lidar variables.

| Land use class | Historical scenario | | | | | | | |
| --- | --- | --- | --- | --- | --- | --- | --- | --- |
| | Fallen CDW | | | | Standing CDW | | | |
| | Predictors | Bias (%) | R² | RMSE (%) | Predictors | Bias (%) | R² | RMSE (%) |
| Intact | **Site Factor, Canopy Relief Ratio,** Radom slope | 0.00 | - | 28.8 | **Gap count,** Gap count² | -8.50 | 0.18 | 56.0 |
| Logging | Age since logging, **Gap area, P05 last returns** | 0.00 | 0.54 | 37.2 | - | - | - | - |
| Burning | Age since fire, Gap area, **number of fires** | 5.4 | 0.43 | 52.2 | Age since fire | -7.0 | 0.11 | 58.0 |

