# Peer review of "Estimation of Coarse Dead Wood Stocks in Intact and Degraded Forests in the Brazilian Amazon Using Airborne Lidar"

_Biogeosciences, 2019_

## Referee Comment (RC1) · Anonymous Referee #1 · 8 Apr 2019

The manuscript by Scaranello et al. deals with a new approach to quantify the CWD in tropical forests. The approach is based on Airborne LiDAR and it may be applied to other forests as well, at least I see no reason why this should not be possible. Based on a return density of at least 13 returns per $m^2$ more than 14,000 ha of Brazilian forest were scanned. Ground-truth was obtained by forest inventory. Additionally, LandSat Imagery was used to include disturbance history as an additional explanatory variable.

The overall quality of the discussion paper is, to my impression, good. I appreciate the concise way of writing and the nicely structured style (well chosen paragraphs). All chapters are in good balance; language is mostly very good and as far as I can tell

[Figure]

**BGD**

references are provided in an appropriate manner and number. To me this is a solid work.

I would have enjoyed to see a more hypothesis-based pre-selection of LiDAR metrics that should be related to CWD, rather than an exploratory analysis of all tested metrics. The authors may at least explain, why they chose the presented metrics out of the myriad of metrics available from LiDAR.

Individual scientific questions/issues: To me, the RMSE are always fairly high. 51.6% (p.10 l.26) is a lot, even though the authors show that earlier models performed weaker even in boreal forest (78.8%). I therefore appreciate that the authors stated that their work may help "reducing uncertainty", rather than selling it as a highly accurate tool.

A few purely technical suggestions:

p.3 l.10: shouldn't it be "losses in aboveground" instead "of"? p.3 l.11: Shouldn't it be "at the forest floor" not "in"? p.6. 9 and 10: Would be nice to refer to Table 1 again, so the reader knows where to find the different metrics p.8 l.3: It becomes obvious what the referee criticized: "CWD (including fallen and standing dead wood". I agree the terminology may be revised as suggest by the other referee. p.8 l.14: coarse woody debris with "y" p.8 l.27: "site-related"

---

## Short Comment (SC1) · 15 May 2019

—

*A note upfront from the submitting person: This review was prepared by Alice Gargano and Marc Grob, both master students in geography at the University of Zurich. The review was part of an exercise during a second semester master level seminar on "the biogeochemistry of plant-soil systems in a changing world", which I organize. We would like to highlight that the depth of scientific knowledge and technical understanding of these reviewers represents that of master students. We enjoyed discussing the manuscript in the seminar, and hope that our comments will be helpful for the authors.*

—-

[Figure]

The study by Scaranello et al. deals with a new approach to quantify coarse dead wood using lidar measurements combined with Landsat images and forest inventory data. The study area consists of 14 test sites and 103 transects with a total length of 48 kilometers in the Brazilian Amazon. The newly suggested approach should make it possible to gather the amount of coarse dead wood in intact, logged, burned or logged and burned forests and help to estimate their importance as carbon stocks.

We appreciate the chosen topic for its originality, which in our opinion justifies conducting the study. It is interesting to see how the problem of quantifying coarse dead wood has been identified and how a new approach can be adopted using remote sensing to complement the information gap. The comparison of the collected remote sensing data with Landsat time series and forest inventory samples seems comprehensible and is well described, despite its technical and interdisciplinary complexity. The structure of the paper is generally well chosen and the language as well as the abbreviations are consistent and of good quality.

Generally, the paper is very interesting to read and contains a detailed description, yet the abundance of numbers in % or meters per each dataset makes it hard for the reader to fully understand and keep track. It is very technical, mainly discusses the lidar-predictor model and has only a limited part involving the dead wood's contribution to carbon storages, which makes us, as non-experts, reflect whether it really belongs to a biogeosciences journal or rather to a remote sensing one. However, we agree with the importance of the topic and think the paper can be published after some modifications.

—- At the beginning of the paper we would have appreciated a short section on the detailed contribution of dead wood to carbon storage and the related processes. It would be a good introduction to the topic and could better justify the importance of the study.

Moreover, the paper does not describe clear hypotheses and expectations regarding data collection and results. This part is particularly important due to the existing uncertainties and unexploredness regarding coarse dead wood. We would have welcomed more information from such a pioneering study on unexplored aspects of the contribution of coarse dead wood to the carbon storage. In contrast, the section on the tested lidar metrics is extremely detailed in our opinion. Although this aspect is important to describe, it can be challenging for remote sensing non-specialists to follow and too technical, we would suggest the addition of figures (for example after line 6 in page 6, guiding figures would help). The overweight of the material and methods section compared to the results makes us understand that this paper works as a preliminary study in a new research field. Due to the limited predictors and their overall performance (page 11, line 23) the question arises whether the study was conducted too early.

Other uncertainties we came across are concerning the justification of the chosen test sites. The Amazonian Forest is extremely large and your study focuses only on a small tract of the its area (according to the stated lidar-data used). How representative is this, given the not very high performance of the predictors?

In the results section, you could add a small table summarizing the results you compare, the best resulting predictors and comparison between lidar-only and historical models. Moreover, in the discussion you accentuate that the differences in site-specific characteristics are uncovered by the slight improvement achieved by the historical model. The RMSE decreased of only 1% making it questionable whether historical scenarios are detailed enough.

—- The following specifications and questions also emerged after reading the paper:

Page 2, line 11: What are the structural variables exactly? Maybe you could give a short definition.

Page 3, line 4: The exact value of 60 Pg-C is in our opinion not appropriate, due to uncertainty. You could either give a range or use "approximately" to reduce the anchoring effect of the absolute value.

Page 3, line 10f.: Too many percentages are reducing the readability.

Page 3, line 25f.: Considering that the study was conducted during three years, the question arises whether any feedback or temporary changes during this period have been considered.

Page 4, line 10: Why did you choose this period? How can that be justified?

Page 4, line 25f.: Were the assumptions for regression met? Normality, homoscedasticity, etc.

Page 7, line 14: Why was the subset selection approach chosen?

Page 10, line 8: What is the single event (here mentioned) exactly?

Page 19, Figure 1: Figure 1 does not really help nor guide the reader, we believe it could be improved by adding an inset map to facilitate the orientation and readability of the location of the test sites and points for each site instead of the abbreviations of the test sites. Furthermore, the canopy height map colours are hard too identify regarding the visual differentiation on the small map as well as the tiny legend. We suggest to either enlarge the map and legend or change at least the colourway. The small figure on the right top shows a rather unnatural pattern between fallen and standing wood, which we cannot explain. The explanation of the used statistical approaches could contain a bit more details, like why the subset selection approach was used and whether all regression assumptions were met.

Page 23, Figure 5: The colourway of the figure is cartographically questionable and the legends have different ranges, making comparison challenging. Furthermore, the predicted mean (red dotted line) and the field-based mean (black dotted line) are hard to detect and not explained in the legend. Moreover, the arrangement of the graphs makes it hard for the reader to compare the different graphs, maybe you use a consistent arrangement for all figures and graphs.

---

## Referee Comment (RC2) · Anonymous Referee #2 · 7 Jun 2019

General:

This is a good paper in that it represents a substantive advance in the mapping of dead wood in tropical forests, using remote sensing data, such that the dead wood estimates can be mapped. The paper also goes so far as to map the estimates. Such methods are needed as standing and downed dead wood carbon stores are a sizeable component of the total carbon loads. The estimates presented also make sense given the disturbance history of these sites and I like that the authors included models that account for disturbance history, in addition to lidar-only models.

Major comments:

I don't like the term "standing CWD" because it sounds like an oxymoron. The terms "standing dead wood" or "fallen dead wood" make sense, but the word "debris" strongly implies that the dead wood has already fallen. Similarly, to say "fallen CWD" sounds redundant. Alternatively, call the "standing CWD" snags, and the "fallen CWD" simply "CWD".

Reading the methods on the spatial layout of the sample plots and transects, I couldn't quite work out whether all the locations sampled were wholly contained within the extent that lidar metrics were calculated. Clarifying the description would help, as would improvements to Fig. 1 to also show the area within which the lidar metrics were calculated.

Fig. 5 caption. The predicted versus observed means are already shown in the figures themselves and do not need to be cluttering up the caption. Just add the +/- standard errors to the figures as well, below the means.

Minor comments:

Introduction

L5. Delete the "and" before "burned", and add a comma after "burned", so this sentence makes sense.

L10. Do you mean to say "tree" or "live" aboveground biomass? Because the CWD pool could be considered part of aboveground biomass.

L13. Here and elsewhere, more commas are advised. If a list, add commas to the item preceding the "and". In this case, commas are advised after "17%", after the first "burning", and before "respectively".

L26. The second "the" should be deleted. So also could be the first "the". L31. Enclose the phrase "and based on a recent comprehensive review (Marchi et al., 2018)" in commas.

Methods

L10. Data "were" not "was".

Next page, L11. The two citations should not be followed by commas before the year published. L20. Either "at" or "with" but not "at with".

Next page, L29. Change "median of age" to "median age".

Discussion

Section 4.2, L15. Need the verb "are" before "similar".

Last paragraph, L7. Need a comma before "and", which separates two complete sentences.

---

## Author Comment (AC1) · 4 Jul 2019

Response to anonymous reviewer #1, "Estimation of Coarse Woody Debris Stocks in Intact and Degraded Forests in the Brazilian Amazon Using Airborne Lidar"

We thank you for their helpful comments. We respond to all concerns and suggestions below. Reviewer comments are quoted in italics followed by our responses.

Reviewer #1.

Comment: I would have enjoyed to see a more hypothesis-based pre-selection of Li-DAR metrics that should be related to CWD, rather than an exploratory analysis of all

tested metrics. The authors may at least explain, why they chose the presented metrics out of the myriad of metrics available from LiDAR.

Response: We used an exploratory analysis (subset selection) for the lidar only models and a hypothesis-based approach for the historical models as discussed on page 7, lines 14-23 of the submitted paper. As requested by the reviewer, we will expand on the discussion of selected metrics in the hypothesis-based approach in the section of page 7 beginning with line 19. Following below is a draft text.

For intact forests, we selected canopy relief ratio as a measure of canopy structure and site factor for aggregating site-specific differences. Previous studies in intact forests suggested differences in CDW stocks, as well as the underlying mechanisms in the CDW input (Rice et al. 2004; Pyle et al. 2008). For logged forests, we selected age since the last disturbance because CDW diminish with time because of decomposition (Chambers et al. 2000). We also selected gap area because tree mortality and CDW stocks were closely related to gap area in intact forests at Tapajós National Forests, Pará (Espírito-Santo et al. 2013). For burned forests, we selected the number of fire events, in addition to age and gap area. A previous study conducted in Paragominas municipality, Para, and Alta Floresta, Mato Grosso, showed the gradual increase of CDW stocks from one to three fire events (Cochrane et al. 1999). In both logged and burned forests we included a measure of forest canopy height correlated to live above-ground biomass (Longo et al. 2016) because aboveground live biomass is significantly correlated with CDW across the Amazon (Chao et al. 2008).

We already have discussed the relevance of variables that were found through the subset selection process. But, as the reviewer suggests, we will provide additional insights to explain the relevance of selected metrics in the exploratory analysis.

Comment: To me, the RMSE are always fairly high. 51.6% (p.10 l.26) is a lot, even though the authors show that earlier models performed weaker even in boreal forest (78.8%). I therefore appreciate that the authors stated that their work may help "reducing uncertainty", rather than selling it as a highly accurate tool.

Response: We agree with the reviewer that a RMSE of 51.6% is not exceptionally good but we note that our models using only lidar data had smaller RMSE (between 33 and 36%). The cited value of line 26 at page 10 is related to the study of Pesonen et al. (2008). We suspect that the reviewer was confused with the RMSE of this study. In a revised manuscript, we would rewrite the sentence to clarify that the 51.6% RMSE belonged to the Pesonen et al., 2008 study as follows, "In the boreal forest the model for fallen coarse woody volume had a relative RMSE of 51.6% and is similar to the performance of the model for burned sites in our historical scenario."

Comment: p.3 l.10: shouldn't it be "losses in aboveground" instead "of"?

Response: We feel that "of" is the proper usage here.

Comment: p.3 l.11: Shouldn't it be "at the forest floor" not "in"?

Response: "At the forest floor" is an acceptable change that we will implement.

Comment: p.6. 9 and 10: Would be nice to refer to Table 1 again, so the reader knows where to find the different metrics

Response: As suggested we will add appropriate references to Table 1 on these three pages.

Comment: p.8 l.3: It becomes obvious what the referee criticized: "CWD (including fallen and standing dead wood". I agree the terminology may be revised as suggest by the other referee.

Response: Please see our response to reviewer #2.

Comment: p.8 l.14: coarse woody debris with "y"

Response: We believe that the descriptive "total coarse dead wood" is more valuable here than a generic independent variable "y". I use therefore "total coarse dead wood"

for our model.

Comment: p.8 l.27: "site-related"

Response: We will add the "-" as correctly suggested by the reviewer.

---

## Author Comment (AC2) · 4 Jul 2019

Response to anonymous reviewer #2, "Estimation of Coarse Woody Debris Stocks in Intact and Degraded Forests in the Brazilian Amazon Using Airborne Lidar"

We thank you for their helpful comments. We respond to all concerns and suggestions below.

Reviewer #2.

Comment: I don't like the term "standing CWD" because it sounds like an oxymoron. The terms "standing dead wood" or "fallen dead wood" make sense, but the word "debris" strongly implies that the dead wood has already fallen. Similarly, to say "fallen CWD" sounds redundant. Alternatively, call the "standing CWD" snags, and the "fallen CWD" simply "CWD".

Response: Both reviewers agree that "debris" connotes fallen wood found at the forest floor. We understand this concern and we will make the following changes throughout the paper including in the title: (1) "woody debris" will be replaced with "dead wood" and will be further described as either "standing" or "fallen"; (2) "coarse woody debris" will be replaced with "coarse dead wood" that will be abbreviated as "CDW."

Comment: Reading the methods on the spatial layout of the sample plots and transects, I couldn't quite work out whether all the locations sampled were wholly contained within the extent that lidar metrics were calculated. Clarifying the description would help, as would improvements to Fig. 1 to also show the area within which the lidar metrics were calculated.

Response: As noted on page 4, lines 9-11, "The airborne lidar data used in this study were collected between 2012 and 2015, covered a total area of 14,870 ha and overlapped with 103 CWD transects (48 km of total length sampled within 6 months of the lidar airborne campaigns)." We will write two sentences to clarify. In the first, we will describe the airborne sampling and in the second we will describe the ground sampling indicating that all ground sampling locations were wholly contained in the airborne lidar areas of interest. We improved Fig. 1 by adding the canopy height model scene from the AND site as an example of the lidar data covering the field transects for sampling standing and fallen dead wood.

Comment: Fig. 5 caption. The predicted versus observed means are already shown in the figures themselves and do not need to be cluttering up the caption. Just add the +/- standard errors to the figures as well, below the means.

Response: We feel that a few more numbers in the caption are easier to understand than a figure cluttered with uncertainty estimates. The figure is already complex. We

prefer to leave the figure and caption as submitted.

Comment: [p. 2] L5. Delete the "and" before "burned", and add a comma after "burned", so this sentence makes sense.

Response: We will implement this suggestion.

Comment: [p. 3] L10. Do you mean to say "tree" or "live" aboveground biomass? Because the CWD pool could be considered part of aboveground biomass.

Response: We will add the word "live" for clarity.

Comment: [p. 3] L13. Here and elsewhere, more commas are advised. If a list, add commas to the item preceding the "and". In this case, commas are advised after "17%", after the first "burning", and before "respectively".

Response: We will implement this suggestion.

Comment: [p. 3] L26. The second "the" should be deleted. So also could be the first "the".

Response: We will delete both uses of "the" referred to in the comment.

Comment: [p. 3] L31. Enclose the phrase "and based on a recent comprehensive review (Marchi et al., 2018)" in commas.

Response: We will implement this suggestion.

Methods Comment: [p. 4] L10. Data "were" not "was".

Response: We will implement this suggestion.

Comment: [p. 5] Next page, L11. The two citations should not be followed by commas before the year published.

Response: We will remove the two commas.

Comment: [p. 5] L20. Either "at" or "with" but not "at with".

Response: We will delete "at."

Comment: [p. 6] Next page, L29. Change "median of age" to "median age".

Response: We will implement this suggestion.

Discussion Comment: [p. 10] Section 4.2, L15. Need the verb "are" before "similar".

Response: In this case, we will add the verb "were" to match the past tense used throughout the paragraph.

Comment: Last paragraph, L7. Need a comma before "and", which separates two complete sentences.

Response: We will add the comma as suggested.
* * *

---

## Author Comment (AC3) · 4 Jul 2019

Response to interactive comment, "Estimation of Coarse Woody Debris Stocks in Intact and Degraded Forests in the Brazilian Amazon Using Airborne Lidar"

We are especially grateful to Alice Gargano and Marc Grob and their professor Michael W. I. Schmidt for their attention and suggestions.

We respond to all concerns and suggestions below.

Interactive comment submitted by Michael W. I. Schmidt

Comment: Generally, the paper is very interesting to read and contains a detailed de-

scription, yet the abundance of numbers in % or meters per each dataset makes it hard for the reader to fully understand and keep track. It is very technical, mainly discusses the lidar predictor model and has only a limited part involving the dead wood's contribution to carbon storages, which makes us, as non-experts, reflect whether it really belongs to a biogeosciences journal or rather to a remote sensing one. However, we agree with the importance of the topic and think the paper can be published after some modifications.

Response: Our paper is technical and we strived to present a large amount of information in a concise manner. We prefer not to add background material that has been published many times elsewhere. One anonymous reviewer praised our presentation. Here is what reviewer 1 wrote, "The overall quality of the discussion paper is, to my impression, good. I appreciate the concise way of writing and the nicely structured style (well chosen paragraphs). All chapters are in good balance; language is mostly very good and as far as I can tell references are provided in an appropriate manner and number." Although this article sounds technical we strongly believe that this study is appropriate for the broad audience of Biogeoscience concerned with the importance of dead wood in forest ecosystems. Our study is novel because it is the most extensive dataset of CWD ever collected in the tropical forest using identical protocols for intact and degraded forests. Moreover, it is the first study that quantifies the relation between forest degradation and dead wood stocks using a unique methodology that combines airborne LiDAR and forest inventory.

Comment: At the beginning of the paper we would have appreciated a short section on the detailed contribution of dead wood to carbon storage and the related processes. It would be a good introduction to the topic and could better justify the importance of the study.

Response: The contribution of CDW on carbon storage and cycle are described in the following sentences of the introduction.

Forest degradation is accelerating the rate of tree mortality across the tropics (Mc-Dowell et al., 2018), leading to severe losses of live aboveground biomass (AGB) (Berenguer et al., 2014; Cochrane, 2003; Longo et al., 2016; Rappaport et al., 2018) and gain of coarse dead wood (CDW) in the forest floor. Aboveground live biomass decreased 35% after logging and 55% after burning + logging in Paragominas Municipality, in the eastern Brazilian Amazon, whereas in Santarem municipality (central Brazilian Amazon) the aboveground live biomass decreased 18%, 17%, 24% after logging, burning, and burning+logging, respectively (Berenguer et al., 2014).

We will highlight the importance of CDW by replacing the previous sentences with the section below.

Forest degradation is accelerating the rate of tree mortality across the tropics (Mc-Dowell et al., 2018), leading to severe losses of live aboveground biomass (AGB) (Berenguer et al., 2014; Cochrane, 2003; Longo et al., 2016; Rappaport et al., 2018). In several areas of the tropics, the AGB decreased dramatically after multiple events of forest degradation (logging, burning, burning and logging). In central and eastern Brazilian Amazon the AGB decreased between 18-24% and 35-55% in Santarém and Paragominas regions, respectively. On the other hand, forest degradation promotes the increase of CDW in the forest floor. The stocks of CDW increase substantially after forest disturbance by logging and fire. For example, fallen CDW stocks increased from 55 Mg ha-1 in intact forest to 75 Mg ha-1 with reduced impact logging, and to almost 110 Mg ha-1 in a conventionally logged forest in Paragominas Municipality (Keller et al., 2004). The importance of CDW is magnified in degraded tropical forests (Alamgir et al., 2016). In degraded forests, CDW stocks can exceed the live aboveground biomass pool (Gerwing, 2002; Palace et al., 2012). Quantifying the spatial and temporal variability of CDW production and decay is therefore critical to constrain the magnitude and timing of carbon emissions from forest degradation or climate anomalies such as droughts (Leitold et al., 2018).

Comment: Moreover, the paper does not describe clear hypotheses and expectations

regarding data collection and results. This part is particularly important due to the existing uncertainties and unexploredness regarding coarse dead wood.

Response: The reviewers are correct that we did not state an explicit hypothesis to be tested. However, our hypothesis is implicit to the goal statement at the end of the introduction. "Using airborne remote sensing data, we developed the first lidar-derived estimates of CWD for intact and degraded tropical forests including areas that have been logged, burned, and fragmented by deforestation for agricultural expansion." The implicit hypothesis is that lidar metrics can be used to model the distribution of dead wood across tropical forest landscapes. We tested this hypothesis through the development of the regression models.

Comment: The Amazonian Forest is extremely large and your study focuses only on a small tract of the its area (according to the stated lidar-data used). How representative is this, given the not very high performance of the predictors?

Response: The study sites are distributed across a broad area of the Brazilian Amazon. They were not chosen based on a systematic or random design so we cannot be certain that they are representative. Testing the representativeness of our sample goes beyond the scope of this paper and presents a challenge for future studies.

Comment: In the results section, you could add a small table summarizing the results you compare, the best resulting predictors and comparison between lidar-only and historical models.

Response: Tables 4 and 5 summarize our tested models and we use the same table format to facilitate comparison of our results. We prefer not to add an extra table with no new information

Comment: Moreover, in the discussion you accentuate that the differences in site-specific characteristics are uncovered by the slight improvement achieved by the historical model. The RMSE decreased of only 1% making it questionable whether historical

scenarios are detailed enough.

Response: We agree with this comment and we acknowledged this comment in the discussion section. The historical data did not improve our models. We did not attempt to develop more detailed historical information because of the huge effort required to process all this information. This goes beyond the scope of this paper.

Comment: Page 2, line 11: What are the structural variables exactly? Maybe you could give a short definition.

Response: We agree that this statement may be confusing. We should have called these "variables quantifying forest structure," and we will change the text in the abstract.

Comment: Page 3, line 4: The exact value of 60 Pg-C is in our opinion not appropriate, due to uncertainty. You could either give a range or use "approximately" to reduce the anchoring effect of the absolute value.

Response: We will say "about 60 Pg-C."

Comment: Page 3, line 10f.: Too many percentages are reducing the readability.

Response: We believe that all these numbers are important and we prefer to leave these numbers in the article.

Comment: Page 3, line 25f.: Considering that the study was conducted during three years, the question arises whether any feedback or temporary changes during this period have been considered.

Response: The lidar and forest inventory collections were nearly simultaneous such that changes in forest structure during the study minimally influenced changes in stocks of CDW. We do not believe that these changes had influence in our models. Each combination of field sample and lidar sample represents a single unit of our sample design and thus useful for fitting our statistical models. We did not use or present any multitemporal comparisons (lidar flights collected at the same location with temporal

variation).

Comment: Page 4, line 10: Why did you choose this period? How can that be justified?

Response: Our airborne lidar campaigns happened between 2012 and 2015 constraining our analysis for this period. We wrote that "airborne lidar data used in this study was collected between 2012 and 2015." There is no justification except that we had funding during that period to collect a substantial amount of data.

Comment: Page 4, line 25f.: Were the assumptions for regression met? Normality, homoscedasticity, etc.

Response: The answer is yes. As we noted on page 7, line 25, "We log transformed (natural log) the response variables when necessary for improved model prediction and error distribution assumptions."

Comment: Page 7, line 14: Why was the subset selection approach chosen?

Response: As noted on page 7, line 14, "we used the subset selection approach to identify the simplest and most informative combination of variables (Andersen et al., 2014; Miller, 1984)."

Comment: Page 10, line 8: What is the single event (here mentioned) exactly?

Response: "A single event" refers to one episode of degradation such as logging or burning within one year. We will change the phrase to "single degradation event" for clarity.

Comment: Page 19, Figure 1: Figure 1 does not really help nor guide the reader, we believe it could be improved by adding an inset map to facilitate the orientation and readability of the location of the test sites and points for each site instead of the abbreviations of the test sites.

Response: We made improvements in figure 1 as requested by reviewer #2. We studied 14 sites as listed in Table 1. Providing inset maps for each site would require a

large amount of space (several pages) and we do not think that the additional information provided by multiple inset maps is necessary to support the conclusions in our paper. We note that detailed site design information including the locations of all lidar coverages and field plots to ~1 m accuracy are provided in our on-line data and meta-data available at https://www.paisagenslidar.cnptia.embrapa.br/webgis/

Comment: Furthermore, the canopy height map colours are hard too identify regarding the visual differentiation on the small map as well as the tiny legend. We suggest to either enlarge the map and legend or change at least the colour way.

Response: We believe this comment refers to figure 5. If the journal allocates space, then the maps could be enlarged. We will consider alternative color bars although the one we use is quite common.

Comment: The small figure on the right top shows a rather unnatural pattern between fallen and standing wood, which we cannot explain.

Response: We do not understand this comment or know to which figure it refers. However, we note that the estimates for fall and standing dead wood components are more uncertain that the estimates for total dead wood.

Comment: The explanation of the used statistical approaches could contain a bit more details, like why the subset selection approach was used and whether all regression assumptions were met.

Response: These comments have been addressed previously.

Comment: Page 23, Figure 5: The colourway of the figure is cartographically questionable and the legends have different ranges, making comparison challenging.

Response: We do not understand what is questionable about the color bars and so we do not know how to respond to this comment. The color bar ranges are deliberately different because we felt that the within-site distribution of dead wood was more interesting to represent than the cross-site comparison that is already summarized by

means and standard deviations.

Comment: Furthermore, the predicted mean (red dotted line) and the field-based mean (black dotted line) are hard to detect and not explained in the legend. Moreover, the arrangement of the graphs makes it hard for the reader to compare the different graphs, maybe you use a consistent arrangement for all figures and graphs.

Response: We will add text to the figure legend to explain the field-based and lidar predicted means. The arrangement of the graphics is done to conserve space. It is possible to provide a uniform arrangement of the graphics but we do not think this is worth the effort or the space it will use. For example, all histograms could be placed to the right of the maps. This would use up far more space than our compact figure and provide no additional information. Many experts on research presentations shun so called "white space."

---

## Author Response (AR1)

To: Associate Editor, Edzo Veldkamp

Thank you for handling this manuscript. We are very pleased that the reviewer's comments and the feedback from the interactive comment were positive and improved the manuscript.

Editor comment: One thing I want you to consider is to formulate the hypothesis for the historical models in the introduction.

Response: We addressed the formulation of the hypothesis for the historical models in the last paragraph of the introduction section, page 4, line 17.

Sincerely,

Marcos A. Scaranello, on behalf of all co-atuhors

**List of all relevant changes**

Page 3, line 10: We highlighted the importance of CDW on carbon storage and cycle as suggested by Alice Gargano, Marc Grob and their professor Michael W. I. Schmidt (interactive comment).

Page 4, line 17: We addressed the formulation of the hypothesis for the historical models in the last paragraph of the introduction section, as suggested by the Editor Edzo Veldkamp.

Page 4, line 31: We clarified that all field samples used were wholly contained in the airborne lidar areas of interest, as requested by Alice Gargano, Marc Grob and their professor Michael W. I. Schmidt.

Page 8, line 2: We detailed a hypothesis-based approach for the historical models, as pointed by Reviewer #1 – "*I would have enjoyed to see a more hypothesis-based pre-selection of LiDAR metrics that should be related to CWD*"

Page 9, line 31: We provided additional insights to explain the relevance of selected metrics in the exploratory analysis performed for the lidar-only scenario, as pointed by Reviewer #1 "*The authors may at least explain, why they chose the presented metrics out of the myriad of metrics available from LiDAR*".

All minor corrections pointed by reviewer #1, reviewer #2 and interactive comment were applied.

[revised manuscript text omitted]